# Establishment of an Efficient and Rapid Regeneration System for a Rare Shrubby Desert Legume *Eremosparton songoricum*

**DOI:** 10.3390/plants12203535

**Published:** 2023-10-11

**Authors:** Siqi Qiao, Pei Jin, Xiaojie Liu, Yuqing Liang, Ruirui Yang, Wenwan Bai, Daoyuan Zhang, Xiaoshuang Li

**Affiliations:** 1State Key Laboratory of Desert and Oasis Ecology, Key Laboratory of Ecological Safety and Sustainable Development in Arid Lands, Xinjiang Institute of Ecology and Geography, Chinese Academy of Sciences, Urumqi 830011, China; qiaosiqi20@mails.ucas.ac.cn (S.Q.); jinpei22@mails.ucas.ac.cn (P.J.); liuxiaojie@ms.xjb.ac.cn (X.L.); lyuqing007@ms.xjb.ac.cn (Y.L.); yangruirui19@mails.ucas.ac.cn (R.Y.); baiwenwan21@mails.ucas.ac.cn (W.B.); zhangdy@ms.xjb.ac.cn (D.Z.); 2University of Chinese Academy of Sciences, Beijing 100049, China; 3Turpan Eremophytes Botanical Garden, Chinese Academy of Sciences, Turpan 838008, China

**Keywords:** desert legume, tissue culture, hormone concentration, regeneration, *Eremosparton songoricum*, callus

## Abstract

*Eremosparton songoricum* (Litv.) Vass. is a rare and extremely drought-tolerant legume shrub that is distributed in Central Asia. *E. songoricum* naturally grows on bare sand and can tolerate multiple extreme environmental conditions. It is a valuable and important plant resource for desertification prevention and environmental protection, as well as a good material for the exploration of stress tolerance mechanisms and excellent tolerant gene mining. However, the regeneration system for *E. songoricum* has not yet been established, which markedly limits the conservation and utilization of this endangered and valuable desert legume. Assimilated branches derived from seedlings were cultured on several MS mediums supplemented with various concentrations of TDZ or 6-BA in different combinations with NAA. The results showed that the most efficient multiplication medium was MS medium supplemented with 0.4 mg/L 6-BA and 0.1 mg/L NAA. The most efficient rooting medium was WPM + 25 g/L sucrose. The highest survival rate (77.8%) of transplantation was achieved when the ratio of sand to vermiculite was 1:1. In addition, the optimal callus induction medium was MS + 30 g/L sucrose + 2 mg/L TDZ + 0.5 mg/L NAA in darkness. The *E. songoricum* callus treated with 100 mM NaCl and 300 mM mannitol on MS medium could be used in proper salt and drought stress treatments in subsequent gene function tests. A rapid and efficient regeneration system for *E. songoricum* that allowed regeneration within 3 months was developed. The protocol will contribute to the conservation and utilization of this rare and endangered desert stress-tolerant species and also provide a fundamental basis for gene functional analysis in *E. songoricum*.

## 1. Introduction

*Eremosparton songoricum* (Litv.) Vass. is a leafless legume shrub distributed in Central Asia, with a fragmented distribution in the mobile sand dunes of the Gurbantunggut Desert of Xinjiang [1]. *E*. *songoricum* is the only species of the genus in China, and is also a Grade II endangered plant in Xinjiang [2]. As a typical representative of the extremely stress-tolerant plants in the desert environment, this species has evolved a variety of resistance techniques to adapt to long-lasting harsh desert environments, including drought, high and low temperatures, and UV radiation [3]. In the wild habitats, its leaves are extremely reduced into assimilated branches to reduce water loss, and its roots grow horizontally to facilitate cloning regeneration so as to optimize water utilization [4]. This species can reproduce sexually, but its main strategy of reproduction is cloning regeneration through the root [5]. Meanwhile, rapid asexual reproduction expands the inhabited area, which can ameliorate the soil and fix the sand, thus helping to protect the habitat. Therefore, *E. songoricum* is a promising plant for desertification control and environmental protection. The majority of studies on *E. songoricum* were focused on the ecology adaptation mechanisms at morphological [1], physiological [4], and biochemical levels [5]. Recently, some molecular works were carried out in this unique species, such as the identification and screening of reliable internal reference genes of RT-qPCR [5], isolation and stress tolerance function analysis of a natural truncated DREB transcription factor *EsDREB2B* [3], the obtention of high quality genome sequences (China National GeneBank, project ID: CNP0002419), and the genome-wide identification of AP2/ERF family genes [6].

Plant tissue culture, also known as in vitro culture or sterile culture, is an important technology for plant regeneration, conservation, and genetic transformation [7,8]. There are three main regeneration pathways in plant tissue culture: one is the test tube cutting pathway, and there are also the embryogenesis pathway and organogenesis pathway. The organogenesis pathway is divided into direct organogenesis and indirect organogenesis, i.e., the direct induction of adventitious buds through explants or the induction of the disintegration of explants into guava tissue, followed by the differentiation of adventitious buds through guava tissue and then the induction of rooting to form regenerated plants. Woody plants generally have a long growth cycle in the wild, and their bodies are rich in phenolic compounds and oxidase. The problem of tissue browning in sterile culture is very common, and may affect the growth of ex vivo tissues and reduce the regeneration efficiency of woody plants. Due to these reasons, the regeneration system establishment of woody plants have become particularly difficult [9,10]. At present, an abundance of woody plants have been successfully cultured, such as *Populus tomentosa* Carr. [11], *Bixa Orellana* L. [12], *Toona ciliate* Roem. [13], *Betula platyphylla* Suk. [14], *Malus sieversii* (Ledeb.) M. Roem. [15], and *Tamarix elongate* Ledeb. [16]. Currently, several in vitro tissue culture protocols have been established for legume plants, such as *Glycine max* (L.) Merr. [17], *Medicago sativa* L. [18], *Medicago truncatula* Gaertn. [19,20], *Vigna unguiculata* L. Walp. [21], and *Robinia pseudoacacia* L. [22], and only very few woody legumes [23] such as *Pongamia pinnata* (L.) Merr. [24], *Cajanus cajan* (L.) Millsp. [25], *Faidherbia albida* (Del.) A. Chev. [26], *Mucuna pruriens* (L.) DC. [27], and *Robinia pseudoacacia* L. [22] have established tissue culture systems.

*E. songoricum* is a good material for excellent gene mining and has great potential in being a model desert legume plant for stress tolerant mechanism study [2,3,4]. However, the absent of a regeneration system of *E. songoricum* markedly limits further mechanism exploration and gene function analysis. Meanwhile, considering the aggravation of global climate change and human activity disturbances, the protection of this important, rare, and endangered species is urgent. Therefore, the establishment of an effective regeneration system in *E. songoricum* is urgently required for germplasm conservation, the establishment of a transformation system, and the discovery and utilization of gene resources. The current study focused on the establishment of an efficient in vitro regeneration protocol for *E. songoricum* using an assimilated branches regeneration system. Different mediums with different plant hormones for assimilated branch proliferation, rooting, and transplantation were systemically investigated, and callus induction and abiotic stress treatments on callus were also evaluated. This study will provide technical support for the establishment of a rapid regeneration system for *E. songoricum* for the conservation and utilization of this endangered and extremely stress-tolerant species, as well as laying the foundation for further function analyses of stress tolerance genes in *E. songoricum*.

## 2. Results

### 2.1. Influence of Growth Regulators on the Assimilated Branches’ Regeneration

Growth regulators have different effects on somatic embryogenesis and axillary shoot regeneration. The regeneration rate of assimilated branch explants was dependent on the auxin/cytokinin ratio. We selected NAA and 6-BA at different ratios for shoot regeneration. After cultivation on the shoot regeneration MS medium, the new lateral shoots were observed with the different growth phenotypes. The formation of fewer dwarf shoots and vitrified callus were observed with the addition of NAA (0.1 mg/L) at different 6-BA concentrations (0.2 mg/L and 0.6 mg/L) into the shoot regeneration medium. The 6-BA at concentrations of 0.3 and 0.5 mg/L was able to induce regeneration shoots, which were separate for dwarf shoots with vitrified calluses and had less shooting with vitrified calluses. In contrast to these, regenerated shoots were vigorous, and calluses formed when the 6-BA concentration was 0.4 mg/L (Figure 1A, Table 1). The 6-BA at concentrations of 0.2 and 0.3 mg/L was able to induce regeneration with an average of 2–3 shoots for one explant, and the average seedling’s height was 1.59 cm at 35 d. There were 2–3 shoots from one explant at the concentration of 0.5 and 0.6 mg/L (Figure 1B). The 6-BA at concentrations of 0.5 mg/L and 0.6 mg/L was able to induce the dwarf seedling with an average plant height of 1.2–1.4 cm (Figure 1B,C). In contrast, after 35 days of cultivation with the regeneration medium supplemented with 0.1 mg/L NAA and 0.4 mg/L 6-BA, the regeneration number of shoots was the highest, up to four new lateral shoots regenerated from the explants (Figure 1B), and the seedling height was the highest with an average height of 1.98 cm (Figure 1C). In general, the shoot regeneration MS medium with 0.4 mg/L 6-BA and 0.1 mg/L NAA was selected as the best medium for the regeneration of *E. songoricum* seedlings.

### 2.2. Effect of Growth Regulators on Assimilated Branch Rooting

To obtain complete plantlets of *E. songoricum*, the growth factors NAA and IBA were selected for root induction (Table 2). Growth phenotype observations showed that young roots were visible at the bottom of seedlings after 15 days of culture. After 30 days of culture, the rooting rate (%) and root length (cm) were calculated (Figure 2A). Collectively, the rooting rate of assimilated branches on the WPM medium (rooting rate 24–65%) was higher than that of the MS medium (rooting rate 4–12%). In addition, the difference in sucrose also affected the rooting rate. The sucrose content (30 g/L) was too high to inhibit rooting and the content (15 g/L) was too low to be nutritious. When the sucrose concentration was 15 and 30 g/L, the rooting rate reached 4–32%. Compared with the sucrose concentration of 25 g/L, the rooting rates were separately 9–65% (Figure 2B). The result showed that higher rooting rates from 35 to 65% was observed when sucrose (25 g/L) was in the WPM medium (Figure 2B(d–f)). For the rooting WPM medium with 25 g/L sucrose, the rooting rate of the assimilated branches on the medium without NAA or IBA was the highest, at 65% (Figure 2B(f)). Furthermore, the rooting rate demonstrated the same pattern root length. The longest root length was in the rooting WPM medium without NAA or IBA, with a length of 9.81 cm ± SE (Figure 2C(f)). Compared with the root lengths of assimilated branches in the MS medium, the root lengths were significantly shorter than those in the WPM medium with or without the addition of NAA and IBA. However, there were no significant differences in the root lengths of assimilated branches in the MS medium with different types of auxin hormone or sucrose content. The lengths were from 4.0 to 7.5 cm (Figure 2C). Therefore, the WPM medium containing sucrose (25 g/L) and free auxin hormone was optimally selected for the rooting culture of assimilated branches.

### 2.3. E. Songoricum Acclimatization and Transplants

For improved hydroponics, the regenerated seedling with roots were exposed to the air of the greenhouse for 7 days. Then, the seedlings were transplanted into three kinds of matrixes. After fifteen days’ transplant and cultivation, the survival rates were analyzed. At 1 day from transplant, the seedlings on the three kinds of matrix showed a green and vibrant appearance. At 15 days from transplant, the seedlings planted on the matrix of sand and vermiculite (1:1, *v*/*v*) exhibited vitality and health phenotypes. However, the seedlings showed a wilting and whitish phenotype both on the matrices of soil–perlite–vermiculite (3:1:1, *v*/*v*/*v*) and with pure sand (Figure 3A). The statistical results showed that the highest survival rate of the transplanted seedlings was 77.8%, when the seedlings were planted in the matrix filled with sand and perlite (1:1, *v*/*v*). The survival rate of seedlings was the lowest at 22.2% in the matrix of soil–perlite–vermiculite (3:1:1, *v*/*v*/*v*). The survival rate of seedlings was slightly higher on the matrix of pure sand than on the matrix of soil–perlite–vermiculite (3:1:1, *v*/*v*/*v*) (Figure 3B). Taken together, the most suitable matrix for *E. songoricum* seedling transplantation should be the matrix of sand and perlite (1:1, *v*/*v*).

### 2.4. Influence of Different Auxin and Cytokinin Concentrations on Callus Initiation

Thirty-day-old in vitro assimilated branches of *E. songoricum* were used as explants to initiate callus induction. To identify the most efficient medium composition for callus induction from explants, different hormone ratios of concentrations were examined (Table 3). Most of the assimilated branches as explants showed callus formation after ten days, producing milky yellow-greenish, compact, and friable calluses from the edges of the explants on medium containing both NAA and TDZ. The high ratio of NAA/TDZ was not conducive to inducing embryonic calluses. The result showed that the phenotypes of the induced callus were white, brown, and friable on the MS mediums containing NAA (1.0 and 2.0 mg/L) and TDZ (0.5 and 1.0 mg/L) (Figure 4A(d,e,g,h)). The induction ratios of calluses were relatively low, approximately from 50% to 68% (Figure 4B(d,e,g,h)). Also, the calluses showed the red, friable, vitrified phenotype at the induction medium with 2 mg/L NAA and 2 mg/L TDZ (Figure 4A(i)). However, when the induction medium contained NAA (0.5 mg/L) and TDZ (2.0 mg/L), the callus formed a milky, yellow-greenish, and fluffy embryonic callus (Figure 4A(c)) and the induction ratio reached 76% (Figure 4B(c)). Taken together, the optimal medium for the induction of embryonic calluses was the MS medium with 30 g/L sucrose, 0.5 mg/L NAA, and 2 mg/L TDZ.

### 2.5. Salinity and Drought Stress Treatments of E. Songoricum Calluses

In order to determine the sensitivity of calluses to salt and drought stresses, 15-day embryonic calluses were placed on the callus induction MS medium supplemented with different concentrations of NaCl and mannitol. After 30 days of treatment, the callus cultured in normal medium (without stress treatment) grew well, with a yellowish phenotype and soft texture. The callus treated with NaCl showed a brownish-red callus, and the texture became compact; similarly, the calluses on the MS medium supplemented with mannitol were yellow, poorly grown and hard (Figure 5A). Compared to the average area of calluses in the control (22.18 ± 1.206 cm^2^), the average area of the calluses under NaCl treatment was significantly reduced, at 16.43 ± 0.715 cm^2^ and 16.2 ± 1.56 cm^2^ under 50 and 75 mM NaCl stress, respectively. Under 100 mM NaCl treatment, the average area of the callus was 14.42 ± 1.145 cm^2^ (Figure 5B). Through the measurement and analysis of the callus area, the average areas of the callus on the medium with 100 mM and 200 mannitol were 19.12 ± 1.7 cm^2^ and 17.66 ± 1.45 cm^2^, separately. When the callus was treated with 300 mM mannitol, the average area of the callus was 15.52 ± 1.5 cm^2^, which was significantly reduced compared with that on the MS medium (Figure 5C). Compared to the average weight of the callus in control (7.71 ± 0.060 g), the average weight of the callus under NaCl treatment was significantly reduced, which an average weight of calluses on 50 and 75 mM NaCl stress, respectively, at 6.18 ± 0.024 g and 4.82 ± 0.038 g. Under 100 mM NaCl treatment, the average weight of the callus was 4.2 ± 0.027 g (Figure 5D). The weight of calluses on mannitol medium data showed that the average weight of the callus was 5.81 ± 0.027 g and 4.43 ± 0.049 g, separately, on the medium with 100 mM and 200 mannitol. When the callus was treated with 300 mM mannitol, the average weight of the callus was 3.7 ± 0.027 g, which was significantly reduced compared with the control (Figure 5E). Moreover, to better quantify whether the callus was stress-induced due to the oxidative status changes, the H_2_O_2_ and proline content was measured in the callus after salt and drought stresses (Figure 6). The general trend of H_2_O_2_ content was to increase with increasing stress intensity (Figure 6A,B). When the callus was treated with 100 mM NaCl, the H_2_O_2_ content increased to 4.44 mmol/gprot (Figure 6A). When the callus was treated with 300 mM mannitol, the H_2_O_2_ content increased to 8.95 mmol/gprot (Figure 6B). Additionally, when the callus was treated with 100 mM NaCl, the proline content was 0.026 μg/g (Figure 6C). When the callus was treated with 300 mM mannitol, the proline content increased to 0.028 μg/g (Figure 6D). Collectively, all the above results indicate that the susceptibility breakpoint of the *E. songoricum* callus was 100 mM NaCl treatment, and that the *E. songoricum* callus treated with 300 mM mannitol on the MS medium could be used as a proper drought stress treatment in subsequent gene function tests.

## 3. Discussion

### 3.1. Specificity of Explant Selection for E. Songoricum

Explants used in different species with success in the regeneration of plants are diverse, and include the leaf, shoot tip, root, seed, embryonic axes, epicotyl, and protoplast [21,22,26,28]. Leaf and hypocotyls have often been used as explants for legume plants’ regeneration system establishment [24,25]. Considering that the leaves are too thick and will have fallen off after 40 days’ growth, and the hypocotyls of *E. songoricum* are too short, the leaves and hypocotyls of *E. songoricum* were not selected as explants in this study. We selected assimilated branches as explants, mainly because they are abundant, easy to obtain, and fast-growing. Based on our results, one seed can produce about fifty assimilated branches after 40 days of culture, each of which can expand to four branches, and finally one seed can produce 200 regenerated plants over a three-month cycle, which proved that assimilated branches are a good and efficient explant choice. This may be due to the strong clonal regeneration ability of assimilated branches, which ensures the possibility of more rapid and efficient regeneration.

### 3.2. The Effects of Plant Growth Regulator Combinations on the Assimilated Branch Regeneration Rate

Plants have different responses to plant growth regulator induction. Research shows that the appropriate combination of cytokinin and auxin is conducive to providing a higher regeneration rate [28]. A combination of cytokinins (such as 6-BA or TDZ) and auxins (such as NAA) is often used to induce shoot regeneration and root formation in plants [29,30]. Our data also showed that the MS with 6-BA and NAA was the best concentration combination. The type and concentration of cytokinins will also affect the shoots’ regeneration rate [31,32], and many species showed a regeneration rate that firstly increased at a relatively lower cytokinin concentration, but was inhibited with higher concentrations, such as in *Aeschynanthus pulcher* (Blume) G. Don. [33], *Sinningia Hybrida* Voss. [34], and *Salvia plebeia* R. Br. [35]. Similarly, in this study, we found that 0.2 and 0.3 mg/L 6-BA were able to induce regeneration with the average two to three branches found in one explant, and the average branch height was 1.59 cm at 35 d; however, when the concentration of 6-BA increased to 0.5–0.6 mg/L, two to three branches were also induced, but the plant height was significantly lower compared with the low 6-BA concentration. Furthermore, the addition of NAA at a concentration of 0.1 mg/L had a significant promoting effect on the regeneration rate of *Bougainvillea buttiana* [32]. Our data also showed that MS with 0.4 mg/L 6-BA and 0.1 mg/L NAA was the best concentration combination, in which the assimilated branch had highest regeneration rate (reaching up to four regenerated branches) with better plant growth (the average height was up to 1.98 cm).

### 3.3. The Effects of Plant Growth Regulator Combinations and Sucrose Concentration on Root Formation

Many studies have shown that WPM has higher rooting rate for woody plants compared with MS medium [15,36], and the rooting of the regenerated shoots was enhanced using WPM supplemented with different growth regulators. Similarly, in this study, the rooting rate of the WPM medium was higher than that of the MS medium. However, we found that the rooting rate in the WPM medium without auxin hormone (up to 65%) was significantly higher than that of the WPM medium containing NAA (40%) and IBA (36%) in *E. songoricum*. Plant culturing normally requires sucrose as the carbon source for cell proliferation and development, and the availability of sucrose in the culture medium has been found to affect somatic embryogenesis in many plant species [37,38,39]. Studies have shown that the growth of roots is influenced by carbon sources in woody legumes [40]; for example, the root growth was reduced when sucrose concentrations were higher than 20 g/L in *Acacia species* [41]. Similarly, in our study, when sucrose concentrations of 25 or 30 g/L were added to the WPM medium without auxin hormone, the rooting rate reached up to 65%, while the highest rooting rate was only 28% when the sucrose concentration was 15 g/L. Therefore, the WPM medium without the auxin hormone and containing sucrose (25 g/L) was selected as the optimum rooting culture for assimilated branches in *E. songoricum*.

### 3.4. The Effect of Different Plant Growth Regulators on Callus Formation

The effect of different plant growth regulators on callus induction was dependent on plant growth regulator type and concentration. Studies show that 2,4-D and NAA are commonly used auxins for inducing calluses [42,43,44]. Moreover, some studies emphasized that the combination of auxin and cytokinin had a better effect on the induction of callus than auxin or cytokinin alone [41,45], and cytokinin TDZ was reported to cause shoot bud induction from calluses in wood legumes such as *Acacia mangium* Willd. [46]. Similarly, in this study, we found that the best callus induction combination for *E. songoricum* was 0.5 mg/L NAA + 2 mg/L TDZ. Previous studies suggested that 2,4-D was not conducive to subsequent callus differentiation and explant status [47,48], and some results showed that the addition of 6-BA could successfully induce differentiation [46,49,50]. Therefore, in this study, we tried different concentrations of 6-BA (0.2, 0.5, 0.8, 1.0, 2.0 mg/L) alone as well as a combination of 6-BA (0.25, 0.5, 0.75, 1.0, 1.25, 1.5 mg/L) with NAA (0.05 mg/L) to induce callus differentiation, but the calluses still could not be differentiated until now. In future work, we will try to add other plant hormones (such as GA3) and different types of auxins (such as IBA) and cytokinins (such as KT or ZT) to induce the callus differentiation of *E. songoricum*.

## 4. Materials and Methods

### 4.1. Plant Material

*E. songoricum* (Litv.) Vass. seeds were harvested from the Gurbantunggut Desert Xinjiang province, *p*. R. China (88°24′67″ E, 45°58′11″ N) [6], dried, and stored in the Conservation and Utilization of Plant Gene Resources, Key Laboratory of Xinjiang, Urumqi, China.

### 4.2. Acquisition of Explants

Seeds were soaked in 98% (*v*/*v*) sulfuric acid for 4 to 5 min to break physical dormancy [10,11]. When small holes appeared in the seed vessels, seeds were transferred into 75% alcohol for 1 min. After sterilization, the seeds were washed with sterilized distilled water 3 times and then soaked in 3% sodium hypochlorite solution for 6 min of sterilization. The seeds were washed with sterilized water 5 times, and quickly planted in basal MS medium (30 g/L sucrose, 6.0 g/L agar, with vitamins and basal salts). The pH of the MS medium was adjusted to 5.8–6.0 using 1 mM NaOH before autoclaving. The medium was sterilized at 121 °C for 15 min. The seeds in the MS medium were cultured at 24 ± 1°C under a 16 h photoperiod with the illumination of 100 μmol m^−2^ s^−1^ light intensity and 45% relative humidity. Plants were maintained through regular subculturing at monthly intervals to provide fresh media with the same composition. The cotyledons of *E. songoricum* were grown after 5-day seed germination, and assimilated branches started to grow at 12 days. The 40-day assimilated branches chosen as explants were cut into 1 cm segments.

### 4.3. Screening of Shoot Induction and Proliferation Medium

To screen for the most optimal medium composition for plant regeneration, the 1 cm apex-portion of 20-day assimilated branches was carefully excised and planted in the proliferative medium with different auxin/cytokinin ratios (Table 1). The constant concentration (0.1 mg/L) of NAA (1-Naphthaleneacetic acid) was used based on the literature, which indicated that many woody plant shoots induced growth at an auxin NAA concentration of 0.1 mg/L [14]. A range of cytokinin 6-BA (6-Benzylaminopurine) concentrations (0.2, 0.3, 0.4, 0.5, 0.6 mg/L) were added to MS basal medium. The number of shoots and seedling height were continuously observed and measured every 5 days until 35 days. The growth conditions were consistent with the explant culture condition.

### 4.4. Rooting of Assimilated Branches

Thirty-five-day-old assimilated branches from in vitro regenerated plants were excised and transferred to MS and WPM (woody plant medium) basal medium supplemented with different concentrations of NAA and IBA (Indole-3-butyric acid) to initiate root formation. A range of sucrose concentrations (15, 25, 30 g/L) were added to the MS basal medium. Two kinds of auxin were added to the MS and WPM medium for the root inducement.

There were three auxin concentration treatments: (1) a concentration of NAA 0.1 mg/L; (2) a concentration of IBA 0.1 mg/L; (3) MS/WPM mediums with neither NAA nor IBA, which served as the control. (Table 2). The incubation conditions were the same as those used for shoot induction. After 30 days, the rooting rate (%) and root length (cm) were recorded. The experiment was repeated five times with five shoots from assimilated branches for each auxin concentration. The growth conditions were consistent with the explant culture condition.

### 4.5. Acclimatization and Transplanting

Sixty-day-old rooted *E. songoricum* seedlings were first hardened by opening the tissue culture bottles for 7 days. The acclimatized seedlings were then removed from the medium and washed carefully with sterilized distilled water to remove the adhered agar. They were immediately transferred to pots containing three kinds of substrates: a mix of autoclaved soil, perlite, and vermiculite (3:1:1, *v*/*v*/*v*); autoclaved pure sand from the desert habitat of *E. songoricum*; and a mix of autoclaved sand and vermiculite (1:1, *v*/*v*) (25 explants for each of the three different substrates). The pots with plantlets were covered with cling film to maintain a high moisture level. The growth conditions were consistent with the explant culture conditions. After 7 days, the film was partially removed. After 10 days, the film was completely removed and the plantlets were watered once every 5 days.

### 4.6. Callus Induction and Stress Treatments

Thirty-day-old assimilated branches of *E. songoricum* were used to initiate callus induction and implement abiotic stress treatments. The assimilated branches were cut into 1-cm-long branch segments with wounds at both ends. The branch segments were placed in a callus induction medium (basal MS medium with 30 g/L sucrose, 6.0 g/L agar, vitamins, and basal salts, pH 5.8). To determine which cytokinin is suitable for callus induction, the medium contained various concentrations of thidiazuron (TDZ) or 6-BA in different combinations. A range of TDZ or 6-BA concentrations (0.1, 0.2, 0.3, 0.4, 0.5, 1.0, and 2.0 mg/L) were added to the MS basal medium. Somatic embryos were also obtained through culturing friable calluses in a liquid medium containing a certain mixing ratio of auxin and cytokine [15]. In order to determine the ratio of auxin and cytokine, the branch segments were placed in a callus induction medium (basal MS medium with 30 g/L sucrose, 6.0 g/L agar, vitamins, and basal salts, pH 5.8) containing NAA and TDZ (Table 3). Ten branch segments were placed in each Petri dish (90 mm in diameter), containing approx. 25 mL medium for induction. In total, five plates were used for each callus induction treatment. In the initial incubation, the branch segments were cultivated in the dark at 24 ± 1 °C for 4 weeks. The callus induction rate (%) was calculated after 30 days of induction.

*E. songoricum* is an extremely drought-tolerant woody plant. The long growth cycle limits the obtention of transgenic plants and a rapid functional evaluation of resistance genes. Therefore, testing the stress tolerance of *E. songoricum* using calluses will contribute to the establishment of a rapid screen system for abiotic stress resistance genes. Fifteen-day-old calluses were placed in different abiotic-stress mediums, which were callus induced MS mediums containing NaCl and mannitol. For salinity treatment, 50, 75, and 100 mM NaCl were added to the callus-induced MS medium. For drought treatment, 100, 200, and 300 mM mannitol were added to the callus-induced MS medium. Ten pieces of callus were placed in each Petri dish (90 mm in diameter), containing approx. 25 mL medium for salinity and drought treatments. The culture conditions were the same as above.

### 4.7. Determination of Physiological Indicators

The H_2_O_2_ and proline (PRO) contents were measured using detection assay kits (Nanjing Jiancheng Bioengineering Institute, Nanjing, China), according to the manufacturer’s instructions.

### 4.8. Statistical Analysis

Statistical analysis was performed using SPSS 26.0 (SPSS Inc., Chicago, IL, USA) software. Three independent biological replicates were performed to ensure the accuracy of the analyses. The data were analyzed using Duncan’s multiple range test followed by one-way ANOVA at the *p* < 0.05 significance level.

## 5. Conclusions

In this study, a rapid and efficient regeneration system of *E. songoricum* was established, as well as a successful induction and stress tolerance evaluation of callus tissues. To our knowledge, this is the first report on the successful establishment of a rapid (from seed to rooting plants within 3 months) and efficient (one seed could produce at least 200 rooting plants after one cycle of regeneration) regeneration system for desert woody legumes via assimilated branches. The details of the regeneration process of *E. songoricum* are summarized as follows (Figure 7): the sterilized seed was cultured on the MS medium, and the seeds sprouted two cotyledons in 3–5 days. The assimilated branches started to grow after 12–15 days. The 40-day plants produced a great number of assimilated branches (about 50) with healthy and vigorous growth, and these were employed in subsequent experiments. Specifically, the assimilated branches with 1 cm apex-portions were used as explants. The optimal shoot induction medium was MS with 30 g/L sucrose, 0.4 mg/L 6-BA, and 0.1 mg/L NAA, and the process took 25 days. For assimilated branch rooting, the optimal conditions were the WPM medium containing 25 g/L sucrose, taking 20 days. After 5 days of acclimatization, the rooting plants were transplanted into the most-suitable matrix (sand and perlite 1:1, *v*/*v*) for natural growth. For the callus induction, the optimal medium was the MS medium with 2 mg/L TDZ, 0.5 mg/L NAA, and 30 g/L sucrose. Our evaluation concluded that the suitable salt and osmotic stress concentrations for calluses were 100 mM NaCl and 300 mM mannitol, respectively.

## Figures and Tables

**Figure 1 plants-12-03535-f001:**
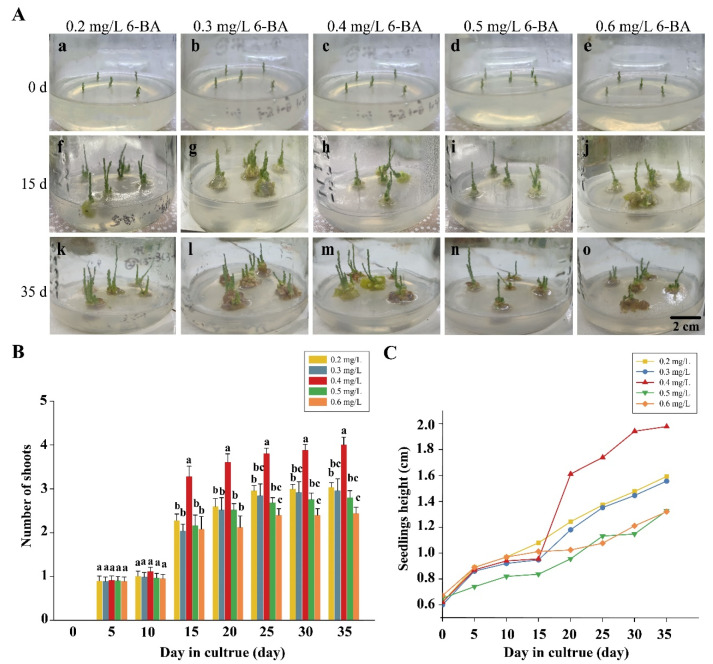
Effect of different mediums with various 6-BA concentrations of assimilated branches. (**A**) The assimilated branch proliferation phenotype observation on different concentrations of 6-BA after 35 days. (**B**) The number of shoots from 0 to 35 days (observed every five days). (**C**) Average seedling height from 0 to 35 days (observed every five days). The average height growth increased abruptly at 15 days with 0.4 mg/L 6-BA added. The NAA content in all shoot regeneration mediums was consistent with a concentration of 0.1 mg/L. The values shown are the means (±SE) of 25 explants. Different letters indicate significant differences among different hormone concentrations from 0 to 35 days. The data were analyzed using Duncan’s multiple range test followed by one-way ANOVA at the *p* < 0.05 significance level.

**Figure 2 plants-12-03535-f002:**
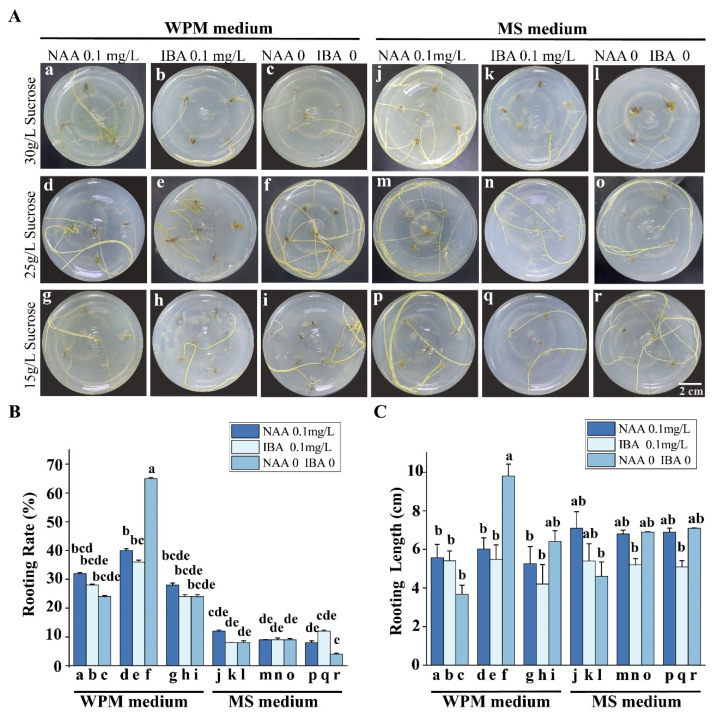
Effects of different mediums on assimilated branch rooting (**A**) Phenotype observation of the roots after 30 days. (**B**) The rooting rate (%) of the assimilations after 30-day culture. (**C**) The statistics of rooting length. Three different sucrose concentrations were selected for MS and WPM medium (30 g/L, 25 g/L, 15 g/L). Three different hormone concentrations were selected for MS and WPM medium (0.1 mg/L NAA, 0.1 mg/L IBA, neither NAA nor IBA). The values shown are the mean (±SE) of 25 explants. Different letters indicate significant differences among different hormone concentrations from 0 to 30 days. The data were analyzed using Duncan’s multiple range test followed by one-way ANOVA at the *p* < 0.05 significance level.

**Figure 3 plants-12-03535-f003:**
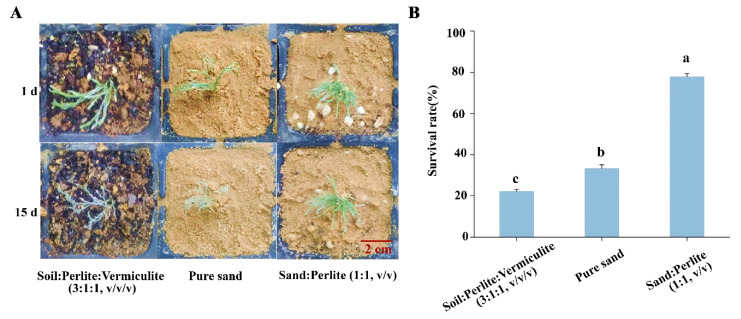
Transplanting of *E. songoricum* on three different matrices. (**A**) Phenotype observation of transplantation of *E. songoricum*. (**B**) Survival rate of seedlings after transplantation. The values show the mean (± SE) of 25 explants. Different letters indicate significant differences among different matrixes, including soil/perlite/vermiculite (3:1:1, *v*/*v*/*v*), pure sand, and sand/perlite (1:1, *v*/*v*) after 15 days from transplantation. The data were analyzed with Duncan’s multiple range test followed by one-way ANOVA at the *p* < 0.05 significance level.

**Figure 4 plants-12-03535-f004:**
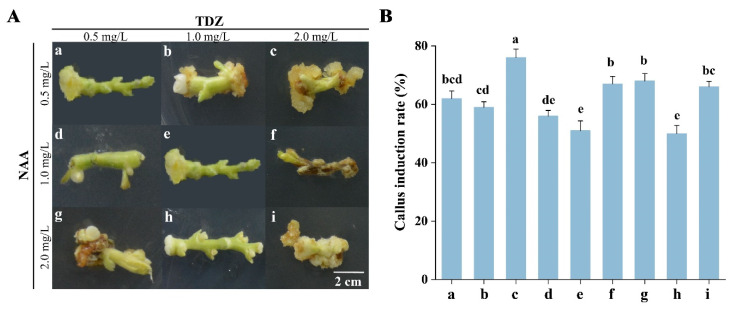
Effect of different hormone concentrations on callus induction of *E. songoricum.* (**A**) Phenotype observation of calluses of *E. songoricum.* (**B**) The induction ratios of calluses. Effect of different hormone concentrations on induced calluses for 30 days. The values showed the mean (±SE) of areas of calluses of 50 explants. Different letters indicate significant differences among different hormone concentrations from 0 to 30 days. The data were analyzed with Duncan’s multiple range test followed by one-way ANOVA at the *p* < 0.05 significance level.

**Figure 5 plants-12-03535-f005:**
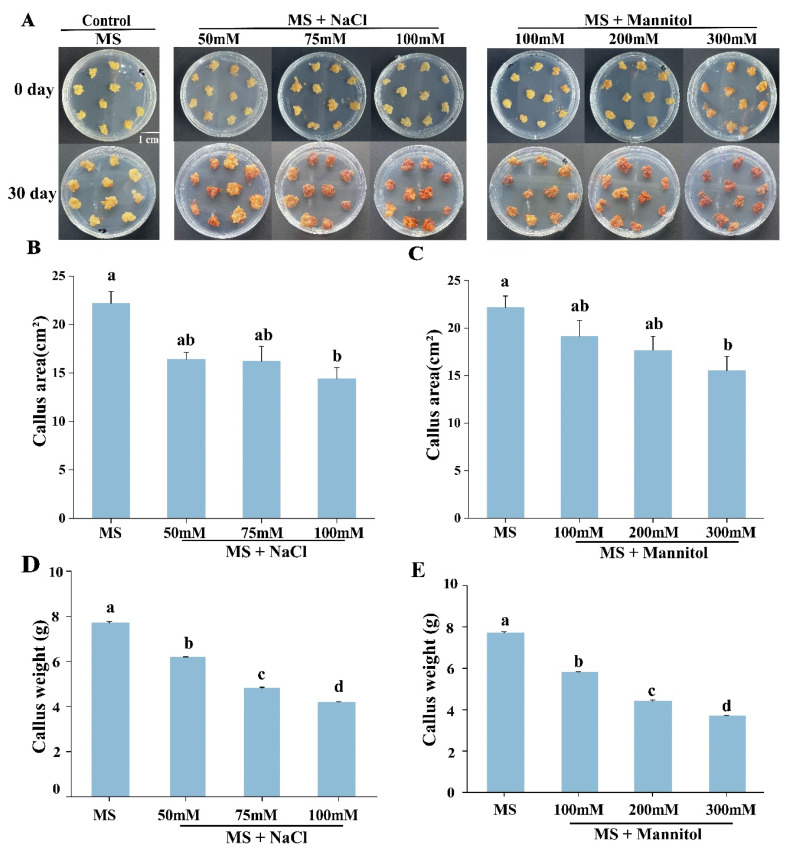
Effect of salt and drought stress treatments on calluses of *E. songoricum.* (**A**) Phenotype observation of calluses on MS medium supplemented with various concentrations of NaCl and mannitol; (**B**) callus area statistics analysis after salinity stress for 30 d on the MS medium supplemented with different concentrations of NaCl (0, 50, 75, 100 mM). (**C**) Callus area statistics analysis after drought stress for 30 d on the MS medium supplemented with different concentrations of mannitol (0, 100, 200, 300 mM). (**D**) Callus weight statistical analysis after salinity stress for 30 d on the MS medium supplemented with different concentrations of NaCl (0, 50, 75, 100 mM). (**E**) Callus weight statistics analysis after drought stress for 30 d on the MS medium supplemented with different concentrations of mannitol (0, 100, 200, 300 mM). The values showed the mean (±SE) of areas/weights of calluses from 50 explants. Different letters indicate significant differences among different mediums. The data were analyzed using Duncan’s multiple range test followed by a one-way ANOVA at the *p* < 0.05 significance level.

**Figure 6 plants-12-03535-f006:**
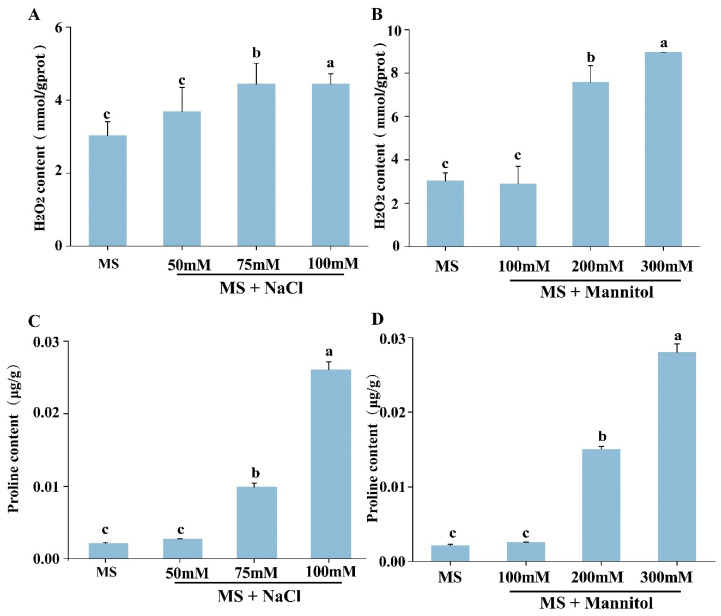
Physiological index changes of calluses under salt and drought stress treatments. (**A**) H_2_O_2_ content of calluses after salinity stress for 30 d on the MS medium supplemented with different concentrations of NaCl (0, 50, 75, 100 mM); (**B**) H_2_O_2_ content of callus after drought stress for 30 d on the MS medium supplemented with different concentrations of mannitol (0, 100, 200, 300 mM); (**C**) proline content of calluses after salinity stress for 30 d on the MS medium supplemented with different concentrations of NaCl (0, 50, 75, 100 mM); (**D**) proline content of calluses after drought stress for 30 d on the MS medium supplemented with different concentrations of mannitol (0, 100, 200, 300 mM). The experiment included three biological replicates; each contained 3 technical replicates. Different letters indicate significant differences among different mediums. The value showed expression level as the mean ± SE (*n* = 9), and data were analyzed using Duncan’s multiple range test followed by a one-way ANOVA at the *p* < 0.05 significance level.

**Figure 7 plants-12-03535-f007:**
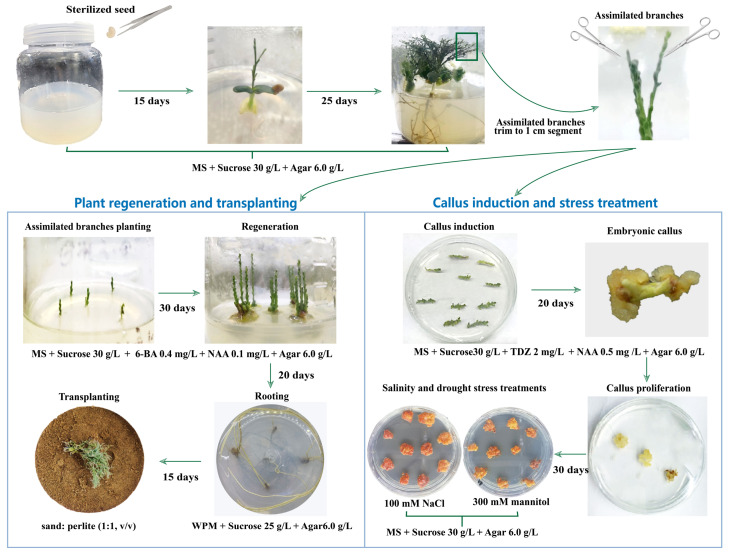
Schematic representation of *E. songoricum*’s rapid regeneration in vitro. *E. songoricum* seeds were sterilized and incubated in normal MS medium. After 40 days of culture, the delicately assimilated branches with 1 cm apex-portions were used as explants for regeneration and callus induction. The optimal mediums and culture times were labeled in each step, and the suitable salt and drought stress concentrations were also provided.

**Table 1 plants-12-03535-t001:** Effect of medium components in combination with MS medium on the shooting of the assimilated branch.

Group	Medium	6-BA (mg/L)	Growing Status
1	MS + 30 g/L Sucrose + 6.0 g/L Agar + 0.1 mg/L NAA	0.2	Less dwarf shooting, vitrified callus formed
2		0.3	Dwarf shooting, vitrified callus
3		0.4	More vigorous shooting, callus formed
4		0.5	Less and dwarf shooting, vitrified callus
5		0.6	Less dwarf shooting, vitrified callus formed

**Table 2 plants-12-03535-t002:** Rooting medium with different sucrose and auxin concentrations.

Group	Medium Components			
	Medium	Sucrose (g/L)	NAA (mg/L)	IBA (mg/L)
a	WPM medium	30	0.1	/
b		30	/	0.1
c		30	/	/
d		25	0.1	/
e		25	/	0.1
f		25	/	/
g		15	0.1	/
h		15	/	0.1
i		15	/	/
j	MS medium	30	0.1	/
k		30	/	0.1
l		30	/	/
m		25	0.1	/
n		25	/	0.1
o		25	/	/
p		15	0.1	/
q		15	/	0.1
r		15	/	/

**Table 3 plants-12-03535-t003:** Callus induction medium with different ratios of TDZ and NAA.

Group	NAA (mg/L)	TDZ (mg/L)	Growing Status
a	0.5	0.5	Green, compact
b	0.5	1	White, compact
c	0.5	2	Milky yellow-greenish, fluffy
d	1	0.5	Brown, friable
e	1	1	Greenish-yellow, compact
f	1	2	Brown, friable
g	2	0.5	Brown, vitrified
h	2	1	Cream white, friable
i	2	2	Red, friable, vitrified

## Data Availability

The datasets used and/or analyzed during the current study are available from the corresponding authors on reasonable request.

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
