# Peer review of "Establishment of an Efficient and Rapid Regeneration System for a Rare Shrubby Desert Legume Eremosparton songoricum"

_plants, 2023, doi:10.3390/plants12203535_

Round 1

Reviewer 1 Report

The paper submitted for review discusses the very important issue of the possibility of the propagation of preserved plants using in vitro cultures. Minor comments have been included in the text as a commentary. The manuscript requires minor corrections. However, the authors presented the results of H2O2 and proline measurements in the material examined, but did not describe the method of these measurements. The M&M chapter should be completed.

Author Response

    • please add the author-classifier to the Latin names

    Response: We checked throughout the manuscript and revised accordingly, please see line 68-75, line289-290.

    • in conclusion, I recommend indicating the reproduction rate, as well as the duration from micropropagation to acclimatization.

    Response: We summarized these data in the conclusions section (line 427-446).

    The most efficient multiplication medium was MS medium supplemented with 0.4 mg/L 6-BA and 0.1 mg/L NAA, and the regeneration number of shoots was the up to 4 (new lateral shoots regenerated from one explant). The most efficient rooting medium was WPM + 25 g/L sucrose ( 65%). The highest survival rate (77.8%) of transplanting was achieved when the ratio of sand to vermiculite was 1:1. The optimal calalus induction medium was MS + 30 g/L sucrose + 2 mg/L TDZ + 0.5 mg /L NAA, and the highest induction ratio reached to 76%.

Reviewer 2 Report

Article Establishment of an Efficient and Rapid Regeneration System

for a Rare Shrubby Desert Legume Eremosparton songoricum. by

Siqi Qiao, Pei Jin, Xiaojie Liu, Yuqing Liang, Rui rui Yang, Wenwan Bai, Daoyuan Zhang and Xiaoshuang Li propose a developed protocol for clonal micropropagation and adaptation of the rare plant E. songoricum.

The manuscript is formatted in accordance with the rules and contains all the necessary sections.

The work leaves a very good impression.

Minor comments: 1) please add the author-classifier to the Latin names, 2) in conclusion, I recommend indicating the reproduction rate, as well as the duration from micropropagation to acclimatization.

Author Response

Reviewer #2:

(1)please add the author-classifier to the Latin names

Response: We check all throughout the manuscript and revised accordingly, please see line 67-74, line 289-290.

(2)in conclusion, I recommend indicating the reproduction rate, as well as the duration from micropropagation to acclimatization.

Response: The most efficient multiplication medium was MS medium supplemented with 0.4 mg/L 6-BA and 0.1 mg/L NAA., and the regeneration number of shoots was the most, up to 4 new lateral shoots regenerated from one explant. The most efficient rooting medium was WPM + 25 g/L sucrose (the highest of 65%). The highest survival rate (77.8%) of transplanting was achieved when the ratio of sand to vermiculite was 1:1. The optimal calalus induction medium was MS + 30 g/L sucrose + 2 mg/L TDZ + 0.5 mg /L NAA, and the highest induction ratio reached to 76%.In summary, this is the first report on the successful establishment of a rapid (from seed to rooting plants within 3 months) and efficient (one seed can obtain at least 200 rooting plants after one cycle of regeneration) regeneration system for desert woody legumes via assimilated branches. We mentioned it in the manuscript (see conclusions, line 430-433).